# Assistance for Target Selection in Mobile Augmented Reality

Vinod Asokan*

University of New Brunswick

Kognitiv Spark

Scott Bateman†

University of New Brunswick

Anthony Tang‡

University of Toronto

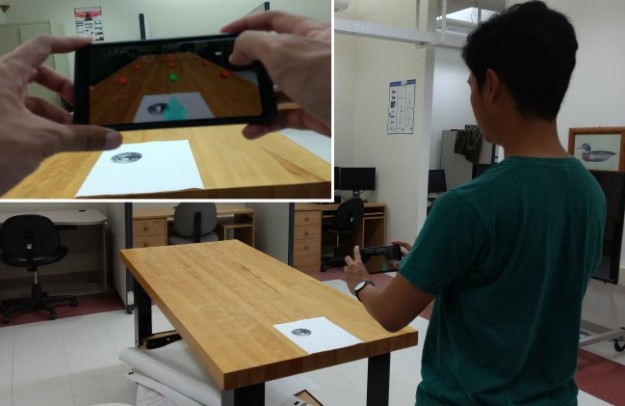

Figure 1. A user viewing our Mobile Augmented Reality system.

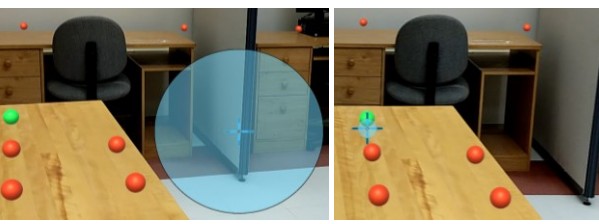

Figure 2. A Bubble Cursor target assistance technique adapted to Mobile Augmented Reality. (Left) The Bubble Cursor expanding towards the nearest target. (Right) The Bubble Cursor shrunk down to only intersect the closest green target.

## ABSTRACT

Mobile augmented reality – where a mobile device is used to view and interact with virtual objects displayed in the real world – is becoming more common. Target selection is the main method of interaction in mobile AR, but is particularly difficult because targets in AR can have challenging characteristics such as being moving or occluded (by digital or real world objects). Because target selection is particularly difficult and error prone in mobile AR, we conduct a comparative study of target assistance techniques. We compared four different cursor-based selection techniques against the standard touch-to-select interaction, finding that a newly adapted Bubble Cursor-based technique performs consistently best across five different target characteristics. Our work provides new findings demonstrating the promise of cursor-based target assistance in mobile AR.

**Keywords**: Mobile augmented reality, target assistance, augmented reality, mobile devices, pointing assistance.

**Index Terms**: Human-centered computing——Human computer interaction (HCI) ——Interaction techniques——Pointing; Human-centered computing——Human computer interaction (HCI) ——Interaction paradigms——Mixed / augmented reality; Human-centered computing——Ubiquitous and mobile computing——Empirical studies in ubiquitous and mobile computing

## 1 INTRODUCTION

With recent advancements in commodity hardware, mobile augmented reality (MAR) is becoming more common. In mobile augmented reality, people use their mobile phones to view and interact with virtual objects placed in the real world. Here, the fundamental task of *selection*, or directing a mobile device in order to locate, point at, and select a visual target, is a fundamental building block for many mobile augmented reality applications.

Target selection is challenging in MAR because targets can vary drastically in terms of location and complexity. For example, UI elements can be located anywhere (on a wall, on a table, behind other objects, behind the user, etc.), have different sizes (by design or because they are located farther away), and they might be moving. Because of these characteristics, a user might need to move their arms, head or body, or otherwise navigate through the physical environment to be able to make selections. Selection tasks are further complicated by the fact that the mobile device cameras have a limited field-of-view.

*e-mail: vasokan@unb.ca
†e-mail: scottb@unb.ca
‡e-mail: tonytang@utoronto.ca

Because target selection is so common in many types of interactive systems, target assistance techniques have long been studied as means to improve general system usability (by improving the speed and accuracy of target acquisition) [24]. Target assistance has most frequently been studies in 2D [5], and it works by dynamically making adjustments to a system to make selecting targets easier (e.g., by moving potential targets closer to the cursor or by dynamically making targets bigger).

In this work, we propose the adaptation of 2D target assistance techniques using a cursor-based selection approach on the mobile device. Here, our approach is different than the typical approach of selecting targets on a mobile screen by directly touching them, we display a cursor for selections (defaulting to the center of the screen); when the cursor rests atop a target, the user can tap anywhere on the screen to commit the selection. Because we make use of a cursor we can apply existing 2D target assistance techniques, which adjust the cursor's position, moving it toward targets with algorithmic assistance, simplifying the on-screen target acquisition task

We adapted three well-known target assistance techniques (Bubble Cursor [16], Sticky Targets [38] and Target Gravity [3]) to explore this approach. The insight is that our approach relies on a 'ray casting' metaphor operating in 2D screen space [21] to do target acquisition in 3D.

To evaluate our approach, we conducted a controlled lab experiment to compare the performance of these newly adapted techniques against a baseline and control condition for MAR target selection. Our system explored each of the selection techniques against five different types of targets (e.g., targets that moved, were hidden, etc.); see Figure 1. Our findings from a 20-participant study show that our adapted Bubble Cursor technique performs best across all target types. We also find that selections through directly touching targets (the conventional MAR selection technique) performs the worst, and that the performance of other assistance techniques is highly dependent on the characteristics of targets.

We make two main contributions in this work. First, we provide evidence that a newly adapted Bubble Cursor technique performs well under a wide variety of target scenarios. Second, we develop a reference for target characteristics that can be carried forward for future study.

## 2 RELATED WORK

Below we briefly survey the foundations of target selection in both 2D and 3D spaces to set the scene for our work.

### 2.1 Target Selection in 2D

Targeting has been widely studied in HCI, and pointing at and selecting onscreen targets is well understood. Fitts' Law [12] states that the index of difficulty (ID), or how difficult a pointing task is, is a function of the distance to the target, and the target's size [24]. This model has been adapted and reinterpreted in several contexts. For example, to better model targeting tasks on mobile touch devices, extensions to Fitts' Law address the use of fingers to select targets [7]. In the context for pointing at targets from a distance on a large screen, [19] determined that ID could be modelled more accurately in distant pointing scenarios by using the angular width of a target and the angular amplitude of the movement to the target (as opposed to using linear distance measures).

#### 2.1.1 Target-Assistance Techniques for 2D

The goal of target-assistance techniques is to enhance the speed and accuracy with which targets are selected. Traditionally, this work has focused on target selection that has occurred on a 2D screen. Balakrishnan argues that these techniques belong to one of three families [5]: techniques that manipulate target size; techniques that manipulate the distance to the target, or techniques that manipulate both size and distance. For example, the Target Gravity technique makes the distance to target smaller, as targets pull the cursor towards them, making acquisition easier [3]. Because a complete review of 2D target-assistance techniques is beyond of the scope this paper, we outline the techniques that we adapt to mobile augmented reality later.

Target assistance techniques have been studied in a wide range of computing environments, including: desktop pointing using a mouse (e.g.,[10][16]), pointing in virtual environments (e.g., [11][14]), and in distant pointing (e.g., [3][15]). Whereas these scenarios rely on indirect interaction techniques, mobile devices rely primarily on direct touch interaction, which means that the target assistance techniques described above are inappropriate. Target assistance techniques, if possible, may be particularly beneficial to mobile touch interfaces, since they suffer from the commonly experienced "fat finger" selection problem [32], where the contact selection point is difficult to predict and the user's fingers visually occlude onscreen targets, making selection error prone.

### 2.2 Target Selection in 3D

Researchers have categorized the actions we perform in 3D virtual environments into four fundamental classes of interaction: selection, manipulation, data input, and navigation [21]. Here, 3D interactions also refer to the interactions made possible in AR and VR, where virtual objects become that targets that users need to interact with. Two relevant approaches exist for selecting targets in a 3D environment [1]: *grasping* and *pointing metaphors*.

In grasping metaphors, a virtual object that represents the user's hand (e.g. a virtual hand or sphere) is positioned in 3D space. This is directly analogous to how we grab physical objects in the real world, by reaching out and grabbing [21].

Pointing metaphors, in contrast, are more like a 2D equivalent of grasping. Here, the user gestures towards the objects to target it. For instance, a ray-casting metaphor is often used, where the ray emanates from the user's hand, finger or pointing device. In many implementations, when the ray intersects with the target, the selection is triggered (e.g., [29]). In other situations, the selection is triggered by a secondary "commit" action, such as lingering over the target, or by pressing a button on a separate controller. The first-generation Microsoft HoloLens device uses a variant of the ray cast metaphor, where a ray is cast from a cursor placed center of device's viewport (accomplished by moving the head). The object selection trigger is completed using an in-air hand gesture that is more easily registered by the device's cameras. The use of the head controlled cursor, provides more precision than could be attained through tracking the hands.

Recent work, has compared grasping using a 6-DoF controller with ray casting using a mouse in AR, and found that ray casting was slower [20]. However, this result might be attributed to the devices used and the fact that all targets were close (i.e., a variety of target characteristics was not evaluated). There is still little work comparing the efficacy of different objection selection approaches in combination with different device scenarios.

Much less work has examined pointing interactions in MAR. However, at least one study by Vincent, et al. [36] examined two different selection techniques using an onscreen cursor in MAR, designed to mitigate the effects of hand jitter. They found that for a 2D object displayed on a real world surface, that a technique displaying a cursor and allowing for fine adjustments using movement of the thumb performed better than a fixed cursor that required movement of the device.

#### 2.2.1 Target-Assistance Techniques in 3D Environments

Target assistance in mixed reality and other non-game specific 3D environments has not yet been studied extensively. Some work has investigated assistance for "grasping". For example, a 3D Bubble Cursor technique works by simplifying target acquisition by dynamically resizing the activation area based on nearby targets [33]. The original 2D Bubble Cursor is described below.

Another approach called BendCast, works for ray cast pointing, bending a visible ray towards targets [30], which effectively decreases the distance to the target. A study of BendCast found it effective, but worked best in scenarios where there were few targets and they were positioned close to the user [9]. Several techniques have proposed casting larger selection areas, effectively increasing target size, to facilitate selection such as the Flashlight ([22]), Aperture Selection ([13]) and Sphere-Casting techniques ([18]). While these techniques all facilitate selections of virtual objects, they either suffer from problems when disambiguating between

multiple potential targets or require additional input interactions to provide this disambiguation.

### 2.2.2 Target Assistance in 3D Games on 2D Screens

Target selection in 3D video games are perhaps most relevant to our scenario of mobile augmented reality. First, many recent video games are played in 3D virtual environments on 2D screens, while making active use of selection techniques (e.g., shooting enemies). First-person shooting video games typically use a gaze-point selection technique, where the camera control and target selection are combined with a selection point (e.g., crosshairs for a gun) affixed to the center of the screen. In 3D games, as with MAR, targets must often be framed within the screen before they can be targeted.

Target assistance techniques are prevalent in many commercial 3D First-Person Shooter games as means to improve performance using different devices or to balance competition between competitors [4][35]. Previous studies have shown that the same target assistance techniques that we study here, among others, also perform well in 3D video games [9][17][23][35].

## 3 DESIGN OF TARGET ASSISTANCE TECHNIQUES FOR MOBILE AUGMENTED REALITY

Target selection is extremely common in mobile augmented reality, and this newer device platform presents some unique challenges that are different from many other forms of 3D interaction. We were interested in understanding how target assistance techniques may apply to generally improve MAR system usability. Our research led us to the idea of applying cursor-based techniques from 2D desktop pointing to this new scenario. However, most mobile interactions do not have a cursor in order to make selections.

To provide a cursor, we borrowed the 'gaze point' metaphor from systems like the HoloLens, 3D video games, and other mobile apps, where a cursor is affixed to the center of the camera view. The use of the cursor for making selections may immediately have benefits when compared to touching targets on the screen in MAR, because it avoids the previously described "Fat Finger" problem. Since an onscreen cursor does not occlude the target and it combines the camera framing and target acquisition tasks that are required in MAR. As we will see, a major benefit of cursor-based approach is that it allows us to apply well-studied target assistance techniques from the literature, many of which work by modifying the cursor's position or selection area.

### 3.1 2D Assistance Techniques Adapted to MAR

Below we briefly describe the three 2D assistance techniques that we identified for use in mobile augmented reality. We selected these three techniques because they represent a range of design alternative used in different target selection scenarios and have all been shown to be extremely effective in terms of improving the speed and accuracy of selecting targets in 2D pointing using a mouse cursor.

*Bubble Cursor:* One of the best performing techniques is Bubble Cursor [16]. Bubble Cursor works by warping an area cursor to the closest target to ensure that a target is always acquired. This adjustment effectively makes targets have zero distance to the cursor and target acquisitions occur by getting "close enough." The Bubble Cursor changes how pointing with a cursor works, relying heavily on visual feedback to help the user understand the relationship between the area cursor and targets.

*Sticky Targets:* The Sticky Targets technique changes the effective width of a target when a mouse-driven cursor passes over top of it [38]. Sticky Targets makes subtle changes to the control-to-display mapping when a cursor is over top of a potential target,

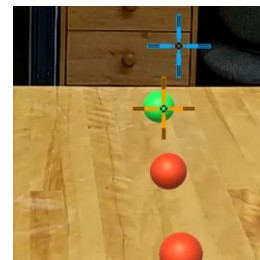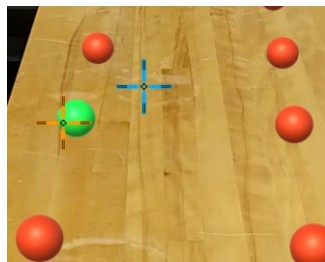

Figure 3. Two Target Assistance Techniques. Orange cursor shows the actual cursor displayed to user, blue cursor is added for illustration, shows the original cursor position before target assistance adjustments. (Left) Target Gravity warps the cursor towards nearby targets (Right) Sticky Targets causes the cursor to stick to a target as it moves over.

giving the illusion that the target feels "sticky," and making a target wider in movement space. The Sticky Targets technique performs well in many 2D pointing scenarios by preventing some forms of overshooting, and it can provide some level assistance and be done in ways that is imperceptible to the user.

*Target Gravity:* The Target Gravity technique changes both the effective motor distance to targets for pointing at targets at a distance (e.g. with a large display) [3]. This technique models all on-screen targets with an attractive force that is proportional to the distance between the on-screen cursor and target, which subtly draws the cursor to the target. Forces from all targets act on the cursor at a given time, so the resultant attractive force is calculated as a vector by summing the force of all targets on the screen. Movement updates to the cursor position are applied by the total gravity effect of all targets.

### 3.2 Assistance Technique Adaptation Details

We designed and studied four cursor-based target selection techniques for comparison against the conventional touch-to-select approach. In the cursor selection techniques, the on-screen cursor defaults to the center of the screen, and based on the target assistance technique used, the cursor may move from this position to capture a target. To make a selection, the user places the cursor over a target, and touches anywhere on the screen to trigger the selection.

Our adapted cursor-based selection techniques are different than many previous implementations of selection techniques in 3D UIs (see [29]) in that they use the intuitive 'ray casting' metaphor, but operate in 2D screen space, making them similar to image-plane pointing techniques [28]. Typically, pointing in 3D environments is accomplished by performing ray casting and collisions are detected in the 3D world space model. Our techniques in contrast operate only in screen space – the pixels displayed on the 2D screen.

Our screen space interaction approach allows us to use assistance techniques, both in 'real' AR scenarios where there is a 3D model for objects in the physical world (i.e., a calculated depth position), and simulated techniques that create the illusion of AR but operate in 2D (e.g., by placing 2D images of 3D objects in proximity to fiducial tags). Further, our approach of projecting targets into 2D space means that after our pre-processing, we can more directly apply any 2D target assistance technique.

For the purpose of comparison, we evaluated three cursor-based target assistance techniques adapted from 2D pointing research, comparing them against a baseline cursor selection technique, and a control, touch-to-select technique. The baseline cursor selection

technique uses a center-of-the-screen cursor, but does not provide any target assistance. The control technique is the conventional touch target selection technique used in most MAR applications.

### 3.2.1 Touch – Control Technique

Touch is the common touch screen interaction technique where the user touches objects on the screen. The center point of the contact surface on the screen is used to determine where the touch occurred; if the touch falls inside the bounds of a target, then the selection is made. This is the conventional MAR approach, which requires two target acquisition sub-tasks to complete the selection: pointing the device to acquire the target in the viewport (i.e. framing), and touching the target on the screen.

### 3.2.2 Baseline – Cursor-Based Selection, No Assistance

Baseline uses a cursor (represented as a crosshair) affixed to the center of the screen. Selections are made by placing the center of crosshair directly over the target and tapping anywhere on the screen to trigger a selection. This technique has a visibility advantage over Touch: the selection point is visible (not hidden by the finger) and always precisely at the point of the cursor. All other target assistance techniques extended this Baseline technique.

### 3.2.3 Bubble Cursor

Our Bubble Cursor technique is based on the well-known technique (described above; see Figure 2), which modifies the selection area of the cursor dynamically. In our adaptation, the cursor is represented as a semi-transparent circle at the center of the screen. The circle expands such that the closest target to the center of the screen is always selected. Targets must be within the view of the camera to be selected. If there are two targets in its area of influence the Bubble Cursor shrinks its width until only the closest target to the center of the screen intersects with the selection circle. The maximum diameter of the selection circle is maintained within the dimensions of the mobile device. Bubble Cursor simplifies the target acquisition task by making the effective width of the target larger and the effective distance to the target smaller.

### 3.2.4 Sticky Targets

Sticky Targets modifies the control-to-display ratio of the cursor when it is over a valid target in mouse pointing, making the target harder to overshoot. In our MAR adapted technique, we determine when the cursor was over a target, and calculate angular movement since the last frame (see Figure 3, right). This distance is used to calculate a new cursor position that is a function of a sticky reduction factor and the distance of the movement. From a user's perspective, the cursor sticks to the target, even if the framing moves slightly. This makes the effective width of the target much wider once the center of the screen has passed over the target once, which prevents overshooting.

### 3.2.5 Target Gravity

Target Gravity was chosen for its good performance in the previous study of 3D FPS games [34]. We adapted Target Gravity by giving all targets within the viewport (frame) of the camera an attractive force on the cursor (see Figure 3, left). This force is proportional to the distance from cursor to target, and the target's relative size in 2D screen space. All forces act on the cursor simultaneously, and movement updates to the cursor position are updated by summing all gravity forces (modeled as vectors) from visible targets. Target Gravity makes the effective travel distance to targets shorter once they are within the viewport.

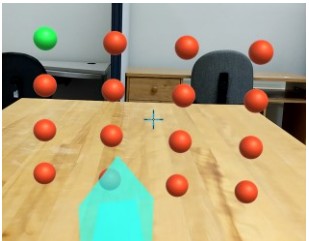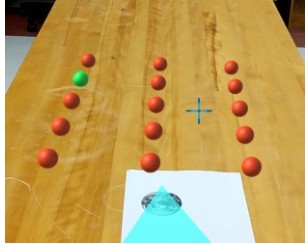

Figure 4. (Left) The UI target arrangement. (Right) The Table Top target arrangement.

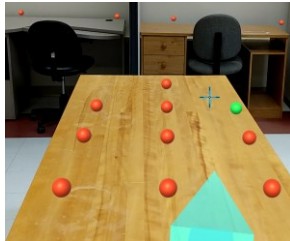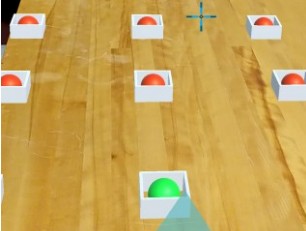

Figure 5. (Left) The Stationary-Distant target arrangement, note targets placed in the distance at the back of the room. (Right) The Obstacle target arrangement.

### 3.3 Target Type Scenarios

To compare our target assistance techniques for MAR, we needed to develop a set of scenarios that were representative of how we expect these targets might appear in real world applications. These needed to mimic a wide range of target arrangements (e.g., coplanar targets; targets that vary in distance from the user; targets that might be obscured by other objects, and so on). While AR applications are not yet commonplace in our day-to-day lives, MAR applications have been common in mobile app stores for many years. We identified target behaviour in the top 20 games and apps in the Google Play store's "Augmented Reality" category (June 2019). For our analysis we installed apps, looked at screenshots and watched videos and created descriptions of the target characteristics. We created short descriptions of the targets observed in each app. The first two authors, then iteratively examined the list and grouped them into categories, assigning labels. This process was relatively quick and straightforward, and resulted in the five basic target arrangements that generalized the target characteristics we observed (described below). We used this list to create arrangements that varied parameters such as distance from the observer, distance between targets, obstacles that could hide or obscure the target, movement of the target and screen space occupied.

Based on our exploration, we found five common targeting scenarios in consumer apps:

- *UI*: Targets are arranged vertically, like a 2D menu interface.
- *Table Top*: Targets are arranged on a horizontal plane, where each target is within 2 meters of the user.
- *Stationary-Distant*: Targets are some distance away from the user. Further targets are visually smaller, which means they are harder to precisely acquire.
- *Obstacle*: Targets are partially visually obscured from the user by some sort of obstacle. To target these, the user often needs to shift their viewpoint – by moving to the side, or up/down – so that they can get accurately acquire the target.

- *Moving*: Targets are moving around in a pseudo-random pattern from a starting point. This mimics the targets in the 3D exploration and game based AR apps.

# 4 STUDY

To determine whether different target assistance techniques can be helpful in MAR, and how their performance might be affected by different target types, we conducted an experiment with 20 participants. Our experiment compared the five different assistance techniques (Baseline, Bubble Cursor, Sticky Targets, Target Gravity and Touch) in five different target type scenarios (Moving, Obstacle, Stationary, Table, and UI).

## 4.1 Apparatus

We developed a mobile augmented reality app using Unity3D. The app implemented the previously described 5 targeting techniques and the 5 target arrangements. The mobile application ran on a Google Pixel 2 XL phone with a 6.2×3 inch display, and Android OS 8.0 with AR Core support. The app automated the running of the study, including the presentation of the different targets and target arrangements. Data logged includes completion time and the number of error (missed target selections before a successful selection) per trial. Completion time was the time from the beginning of a trial and target was displayed for selection and the time when the selection was successfully made. Error rate was calculated as the number of trials with an error divided by the total number of trails.

### 4.1.1 Target Selection Techniques

We used all five target selection techniques described above: three cursor-based target assistance techniques (Bubble Cursor, Sticky Targets, Target Gravity), a baseline cursor-based technique (Baseline), and a control Touch technique.

We predetermined the attraction levels for the Target Gravity and Sticky Targets techniques through piloting, where we selected the best performing values from a reasonable range that test from low to high values of each effect (following the procedure in [3]).

### 4.1.2 Target Type Scenarios

To make the target arrangements somewhat comparable, every target arrangement consisted of 16 targets represented as spheres of size .05m (selected as a target size that allowed the UI, Table Top arrangements to fit comfortably on the screen). While the spheres themselves were the same size, apparent size would change depending on the user's distance from them.

*UI.* A 4×4 grid of vertically arranged targets within reach of the user. Each target was a sphere with a radius of .05m. The closest targets were placed uniformly 0.54m away from the user. In this arrangement the targets were arranged vertically, starting from 0m from the QR code (table) to 0.15m above. See Figure 4, left.

*Table Top.* The targets were arranged close to one another forming a rectangle made up of 5 rows of targets with 3 columns. The targets were spheres of radius 0.05m, the nearest target was 0.4m away from the user and the farthest was 1.1m away. See Figure 4, right. Note the smaller number of targets 15 in Table Top was done to allow the 3 columns to fit comfortably on table tops of different sizes (other Target Types contained 16 targets).

*Stationary – Distant.* The targets were spheres of radius 0.05m, the nearest target was 0.33m away from the user and the farthest was 4m away, the targets were arranged to fill the room. See Figure 5, left.

*Obstacle.* This arrangement is similar to Stationary, but the targets are surrounded by a virtual box that prevents selection.

Targets are placed at varying depths, requiring users to change positions to get nearer and/or to move up/down to change the viewing angle. The targets were spheres of radius 0.05m, the nearest target was 0.33m away from the user and the farthest was 4m away, the targets were arranged to fill the room. See Figure 5, right.

*Moving.* These targets are arranged the same as stationary targets and on becoming selectable the target moves pseudo-randomly around its starting point. The targets were spheres of radius 0.05m, the nearest target was 0.33m away from the user and the farthest was 4m away, the targets had a movement range of 0.5m around its position. Targets moved at approximately 0.25 m/s.

## 4.2 Participants

We recruited a total of 20 participants (3 identified as female, 16 identified as male and 1 person preferred not to say), all were students aged between 20 to 30 (mean: 23.8, sd: 3.64) and all had normal or corrected-to-normal eyesight. With regard to video game experience, 3 participants reported spending more than 7 hours per week on video games, while the remaining 17 played in moderation (i.e. fewer than 5 hours per week). With regard to augmented reality experience, 16 participants reported having experienced augmented reality (mobile and other forms); of these, 4 reported spending up to three hours a week using AR or VR apps/games, and 1 had experience creating a mobile AR app.

## 4.3 Experimental Task

The experimental task was to find and select a series of green colored targets as "quickly and accurately as possible" using the MAR experimental system. Since we were mainly interested in the performance of the different targeting techniques under the different target arrangement scenarios, we provided a guide for participants. The system displayed a 3D blue arrow that points in the direction of the next target to be selected. Upon a successful selection, the target deselects to become red in color and the next target is chosen at random from the 16 targets and is colored green. The blue arrow marker turns towards the new target.

A QR code was kept on the table and before each block of trails and the participant was asked to scan it with the mobile camera to provide the most consistent anchoring of the targets and room as possible. Participants were free to move around through the room to facilitate selection, but were told that "doing so may take more time". Movement was allowed because in some target type scenarios the 'obstacle' targets were difficult to select unless a user was close by, and we expect this to be a common feature of most AR apps. However, our instruction was to prioritize rapid targeting and selection, over moving to a position to make selection extremely easy but requiring more time. Before starting the next Target Type-Technique block participants were asked to return to the same starting position, scanning the QR code anchor.

## 4.4 Procedure

When participants arrived, they were explained the procedure of the experiment, given an informed consent form and they completed a demographics questionnaire. Before starting the experimental trials, participants were introduced to each technique by the experimenter and given an opportunity to try the technique for as long as they needed to feel comfortable.

Once ready, participants proceeded with the experiment working with one of the 5 techniques at a time. Each round consisted of selecting 16 targets, within each of the 5 target arrangements. In total each participant made 480 (5 selection techniques × 5 target arrangements × 16 targets) target selections over the course of the

experiment. The presentation of selection technique is balanced by ordering the conditions in 5x5 Latin Square. The presentation of target type scenario was randomized as the individual targets were randomly selected as the next target, such that all targets were presented (note since the Table Top scenario only had 15 targets, one target was selected twice). Participants completed all Target Type scenarios with a single technique before moving on to the next technique.

After working with each technique and before proceeding to the next, participants were given a questionnaire to collect subjective data on the last technique they had used. Upon completion of the experiment, participants were given a final questionnaire soliciting opinions on their overall experience. The experiment required a total of ~50 minutes to complete.

### 4.5 Analysis

Our main analysis consisted of 5×5 (Technique by Target Type) repeated measures design. Our analysis of subjective data from questionnaires, only compares the effect of Technique on participant responses. Dependent variables were completion time, error rate, NASA TLX score, and agreement with "I would like to use this technique". Subjective data used a 7-point Likert-type scale. We converted the NASA TLX to the 7-point Likert-scale questions to maintain consistency, and following common practice in HCI [26]. We exemplify and support our findings using free-form responses to the post-condition and post-experiment questionnaires.

RM-ANOVA was used to analyze the performance data (when assumptions were met) and post-hoc test used Bonferoni corrections. When the assumption of sphericity was violated, the Greehouse-Geisser correction to the degrees of freedom was used. Subjective data or when the assumption of normality was violated used the Friedman test, and Conover's post hoc test with Bonferoni corrections. For interaction effects, we report pairwise differences for techniques within each target arrangement only, since comparison of techniques between target arrangements is difficult to interpret.

Prior to analysis outliers were removed were removed from the dataset where completion time was greater than 3 s.d. from the grand mean across all participants, which resulted in the removal of 117 of 9800 total trials (~1.2%).

## 5 RESULTS

### 5.1 Completion Time

There was main effect of *selection technique* on completion time ($F_{2.0,39.1}$ = 42.95, $p<.001$), see Figure 6. Pairwise comparisons showed that Bubble Cursor was significantly faster than all other techniques ($p<0.001$), and no other differences was significant.

There was a main effect of *target type* on completion time ($F_{4,76}$=196.3, $p<.001$), see Figure 6. Pairwise comparison showed that all target type pairs were significantly different from one another, with the exception of moving and obstacle.

There was an interaction effect between Selection Technique and Target Type on completion time ($F_{6,4,122.0}$=14.3, $p<.001$), see Figure 6. Within Target Type, the pairwise technique differences were as follows:

- For Moving target types, Bubble Cursor and Target Gravity were significantly faster than all other technique. ($p<.001$).
- For Obstacle target types, all differences were significant except for the Target Gravity-Touch, Target Gravity-Baseline and Baseline-Sticky Targets pairs ($p < .05$).

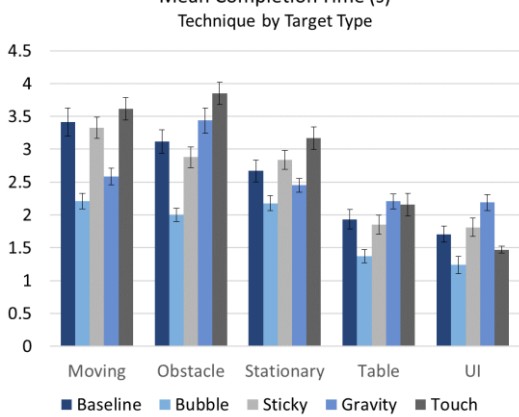

Figure 6. Mean Completion Time in seconds (±1 SE) for Techniques (colours) grouped by Target Type.

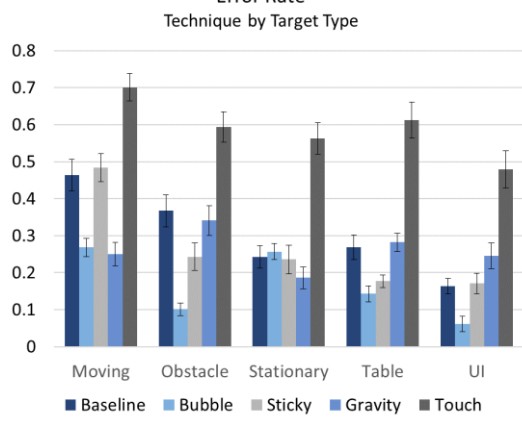

Figure 7. Error Rate (±1 SE) for Techniques (colours) grouped by Target Type.

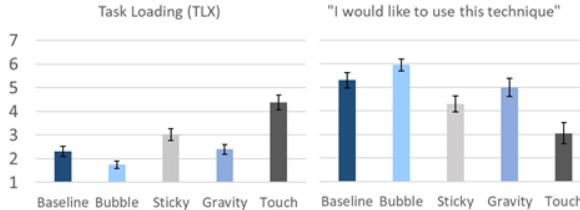

Figure 8. Mean subjective ratings (7-point scale, ±1 SE): Left: NASA TLX (lower is better), Right: agreement with the statement: "I would like to use this technique" (higher is better).

- For Stationary targets, Bubble Cursor was significantly faster than Sticky and Touch, and Target Gravity was significantly faster than touch ($p < .001$).
- For Table Top targets, Bubble Cursor was faster than all other targets ($p<.05$).
- For UI Targets, Bubble Cursor was faster than Target Gravity, Sticky Targets, and Touch, and Touch was faster than Target Gravity ($p<.05$).

## 5.2 Error Rate

There was main effect of *selection technique* on error rate ($F_{4,76} = 113.3$, $p<.001$), see Figure 7. Pairwise comparisons showed that Bubble Cursor had a significantly lower error rate than all other techniques ($p<0.001$), and that touch was significantly higher than all other techniques ($p<0.001$); no other differences were observed.

There was a main effect of *target type* on error rate ($F_{4,76}=28.5$, $p<.001$), see Figure 7. Pairwise comparison showed that all target type pairs were significantly different from one another ($p<.05$), with the exception of stationary-obstacle, stationary-table, and obstacle-table.

There was an interaction effect between *selection technique and target type* on completion time ($F_{7,2,136.7}=6.3$, $p<.001$), see Figure 7. The within target type, pairwise technique differences were as follows for error rate:

- For Moving target types, all pairs were significantly different, except for the Baseline-Sticky Target and Bubble Cursor-Target Gravity pairs. ($p<.001$).
- For Obstacle target types, all pairs were significantly different, except for the Baseline-Sticky Target, Sticky Target-Bubble Cursor, Baseline-Target Gravity and Bubble Cursor-Target Gravity pairs. ($p<.001$).
- For Stationary targets, Touch had a significantly higher error rate than all other techniques ($p < .001$).
- For Table Top targets, Touch had a significantly higher error rate than all other techniques ($p < .001$).
- For UI Targets, Touch had a significantly higher error rate than all other techniques, and Target Gravity had a significantly higher error rate than all other techniques ($p < .01$).

## 5.3 Subjective Results

### 5.3.1 Task-Loading: NASA TLX Score

There was an effect of *Technique* on TLX score ($\chi^2=50.1$, $p<.001$, $df=4$), see Figure 8. Touch had higher task loading than all other techniques ($p<.001$), and Bubble Cursor was lower than Sticky Target ($p<.001$), there were no other differences.

### 5.3.2 I would like to use this technique

There was an effect of *Technique* on agreement with the statement, "I would like to use this technique," ($\chi^2=34.7$, $p<.001$, $df=4$) , see Figure 8. Post-hoc tests showed that all techniques were significantly preferred to touch, and bubble was preferred to sticky ($p<.01$).

## 5.4 Movement Observations

Participants did not move around frequently. Recall, our instruction was designed to coach them to move only when necessary. Further, between condition blocks, we asked participants to return to the same starting spot. While we did not record frequency of movement, we can report with confidence that participants did not feel it necessary to move in most of the conditions. The notable exceptions were the Obstacle and Stationary-Distant scenarios. In the cases, where people moved they reported that it was to because they were having trouble selecting a target, or because they anticipated having trouble selecting a target (due to occlusion or due to a target being small from being located far away).

## 5.5 Summary of Findings

Our study provides the following four main findings:

- Our adapted Bubble Cursor technique performed best across all Target Type conditions and was rated as having the lowest task loading and was most preferred by participants.
- Target Gravity did not perform consistently well, performing best in Target Type conditions with sparsely positioned targets.
- Sticky Targets did not provide any clear benefit above the Baseline technique, and was rated second worst overall.
- The Touch technique, which is the most common mobile selection technique, performed consistently worse across all Target Type conditions, with the exception of UI. Participants also rated it as the worst condition due the difficulties experienced in using it.

## 6 DISCUSSION

### 6.1 Why did Bubble Cursor work best?

The results of our study show that our adapted Bubble Cursor technique worked consistently best across all target type conditions in terms of both time and errors. It seems that the expanded activation area that Bubble Cursor (a large area) facilitated selection. When participants were able to get a target within a frame, with the other techniques they would still need to go through a process of acquiring the target with their finger (for Touch) or with the cursor (for the other techniques). For Bubble Cursor this was much easier, since the closest target would be selected almost immediately within the framing movement. While there might have been other distractor targets closer, we found that participants were able to employ a 'fast and sloppy' interaction style with Bubble Cursor that allowed them to confidently and rapidly select targets. This held true consistently across all target types. Participants appreciated the ease that Bubble Cursor afforded them, which was reflected in subjective responses. Participants rated Bubble Cursor lowest in terms of task loading and as the technique they would most like to use in MAR apps.

One potentially detrimental feature of Bubble Cursor is that it requires some degree of visual feedback. Bubble Cursor needs to provide a visual connection between the cursor and the closest target. This provides the user with the necessary feedback to understand and predict how their movements can influence selection of the next target. However, not all app designs might work well with an additional very visible and potentially distracting visual feature just to support target selection. Future work should investigate the range of design alternatives that can make Bubble Cursor subtler, while still maintain its performance.

### 6.2 Why did Target Gravity only work well for Moving and Stationary-Distant Targets?

Target Gravity is a technique that is commonly used in video games, which we see as being a close analog for target selection in MAR. Target Gravity seemed to work best for Target Types where the targets were sparsely distributed (i.e., Stationary-Distant) or Moving. With these types of targets, the strong attractive force amplifies as participants pursue a target, which facilitates targeting. In these types of targeting scenarios, it seems Target Gravity may be an appropriate technique, especially since it is less visually obtrusive than Bubble Cursor, which uses a large visual representation to illustrate its area cursor. Here, scenarios like games with sparse moving targets seem like promising applications for Target Gravity.

Conversely, Target Gravity performed poorly in the target scenarios with more tightly clustered targets, as in UI and Table. Target Gravity had particular problem with these scenarios, because the Gravity cursor would have to pass by the distractor

targets on the way to making a final selection. However, as the participant passed these targets the Gravity would drag the cursor in the direction of closest target, this was distracting and slightly disorienting for participants.

While target gravity has performed extremely well in some contexts [1][3][4], previous studies have also shown difficulty when other targets are located close by [3]. The original formulation of a Target Gravity technique, by Ahlstrom, et al. [1] called "force field", proposed limiting gravity to a small area around each target. This would mean that the gravity effect would only provide assistance when close to the target (i.e., during the corrective phase and not the ballistic phase). While this would reduce the likelihood of adversely being effected by other targets, it might also mean that the benefit of the technique is largely lost in scenarios with sparse targets. In our future work, we will consider exploring alternative formulations of Target Gravity, to identify whether it can more uniformly provide benefits across a range of targeting scenarios.

We believe designers of MAR apps can safely employ Target Gravity on UI scenarios, since the relatively sparse and buttons in UIs would allow Target Gravity to facilitate selection without having to introduce any additional visual features, like Bubble Cursor, making it easy to learn.

### 6.3 Why did Sticky Targets not work well?

The idea with this technique is to make it easier to stay on a target once it has been acquired, and to prevent sliding off the target or overshooting with the acquisition movement when finalizing the selection. Our findings suggest that the final acquisition of the target may not be a problem in MAR scenarios. However, our observations and the subjective feedback suggest limitations with Sticky Targets, similar to those seen with Target Gravity. Like Target Gravity, Sticky Targets has performed well in a wide range of contexts [4][27][38]; however, Sticky Targets has been shown to have sharp decreases in performance when there are many other targets [3]. When over non-target objects, the Sticky Target effect is still engaged, and the cursor would stick to the targets. When the cursor leaves a "Sticky Target" it returns back to its original position (i.e., the center of the screen) rapidly, causing a jumping effect. Participants mentioned that they found this effect a little jarring and distracting, when it occurred, and this also likely led to cancelling out any of the benefit the technique had (this has also been noted as a detrimental feature in other studies [3]). We believe that Sticky Targets may still have some benefit, but should be used in more limited contexts. For example, applying Sticky Targets to a few highly important targets that need to be acquired more quickly might make sense in some gaming contexts. Its selective application might help to avoid annoying or distracting interactions with other objects.

### 6.4 Why did Touch perform so poorly?

Touch had a significantly higher error rate than all other techniques in all target type scenarios (trial error percentages ranged from ~48% to ~70%). We initially conducted this experiment based on our own frustrations with selecting targets in our own MAR apps that were based on Touch interactions. We see three main reasons for this: first, the targets in our study were small; second, we see it as an instance of the Fat Finger problem; third, our implementation may have been too simple. Participants had difficulties with the imprecise nature of touching a relatively small virtual object, particularly when their fingers obscured the targets, and this led to many missed targets. We consider the challenge of small targets in MAR apps in the next section, below.

Our Touch implementation was the simplest implementation, and followed directly from the Unity (the technology used for building our system) documentation. This involved transforming the touch point from screen space to the near clipping plane in world space, and casting a ray to the far clipping plane. While this equates to the most common approach for target selection on touch interfaces [37], previous work has established that other properties (such as the contact area, touch orientation, among others) can improved the precision of touch-based interactions [6][37]. Future work, should explore improved calculations of touch points for selecting targets in MAR.

While Touch performed poorly in most Target types, it did perform relatively well in terms of completion times for the UI targets. Here, we believe Touch was most like a standard mobile interface. Participants could easily find an initial framing, and maintain it while simply touching the various targets, similar to a standard mobile interface. Despite this, the error rate remained high in the UI condition. While a designer might be tempted to use this result as evidence for using touch as a technique in MAR interface, we would argue that the high error rate might suggest that the Baseline technique, which provides a cursor on the screen but not assistance, would be better, if Target Gravity was not an option.

### 6.5 Why were the error rates high for all techniques?

All techniques suffered from relatively high error rates. Only Target Gravity averaged below a 10% selection errors in just one of the 5 target scenarios. All other error percentages ranged between ~10% to ~34%. From our observations and conversations with participant, we attribute this to the small-sized targets. In particular, most general-purpose mobile interfaces use toolkits that are carefully built to provide widgets and controls that are easy to acquire and interact with (all having a minimal physical size and spacing). MAR does not have an analogous way to ensure that potential targets for interaction have an appropriate size, which can help minimize selection errors. Because virtual objects are placed in the real world, sizing objects for interaction adds an additional level of complexity. Just because a virtual object has realistic dimensions and positioning, does not necessarily mean that it will be easy to point at. Our results demonstrate, though, that target assistance techniques applied atop cursor-based interaction can help address this challenge for MAR apps by drastically reducing error rates and reducing selection times.

### 6.6 Are the different Target Types Scenarios useful?

An important feature of our study was the different Target Type Scenarios, which proved useful to highlight the differences between different techniques. The tasks were designed to strike a balance between internal and external validity; providing good experimental control, while representing a range of target behaviours that come from our observations of a set of commercial MAR apps.

The fact that there was an interaction effect between Selection Technique and Target Type evidences their utility to at least some degree. We believe our Target Type Scenarios at the very least provide a good starting point for other researchers to use in their own studies on target selection and interactions in Augmented Reality.

While we believe that are Target Type Scenarios are representative of some of the most common targets currently represented in MAR, we do believe there are other conceptualizations of the space that are possible. The results of our study and the utility of our Target Types to our study, show that this type of analysis can lead to improved experimental practices. In our future work we plan to reconduct our study by collecting a

larger sample of apps that is more representative of current MAR practices, and by employing a more formal analysis process.

Importantly, however, more work needs to be done to better understand all of the factors that might influence performance, including both additional target properties (e.g., target density, shape, etc.) and device characteristics (e.g., screen size, field-of-view, etc.).

## 6.7 What other ways can MAR selections be assisted?

Target selection is an extremely common task in most modern interactive systems: clicking on a button in a GUI, selecting a restaurant on a map, or shooting an enemy in a video game. In almost all cases, a selection can be described as containing two subtasks: target acquisition (or targeting), and trigger activation [21]. In targeting, a virtual object is indicated for selection (e.g., by pointing at it). In trigger activation, some means of finalizing the selection is committed (e.g., by pressing a button or using a voice command).

The target acquisition subtask (described above) contains two phases: a ballistic phase and a corrective phase [5]. The ballistic phase is the a rapid, open-loop initial movement towards the target, while the corrective phase is a slower, careful, closed-loop movement to acquire the target.

We propose that target selection in MAR has an additional component that is not represented in current characterizations of pointing with conventional systems. In MAR, before the target acquisition phase (described above) a user needs to first do *framing,* which involves orienting the camera of the device toward the target (which could be behind the user or occluded), ensuring that it is properly acquired in the camera frame. This allows the virtual target to be visible, so that it can be selected (i.e., if a target is not visible in the environment, it cannot be selected). Our work, does not address the framing phase, but we present it here to assist in our discussion of the results. Techniques that assist in framing in MAR work by providing visual cues to the location of off-screen targets by (e.g.,[31]). Future work, might explore the design of target assistance techniques that support both phases of target selection in MAR.

## 7 Conclusion and Future Work

In this work, we have described the results a first study of Target Assistance techniques to address the problems with selecting targets in mobile augmented reality. Selecting targets in MAR is particularly difficult because targets can have many varied properties (including size, movement and being occluded), and due to the classic fat finger problem that exists with touch. Our study shows that a cursor-based selection approach combined with target assistance techniques can address these challenges by substantially speeding up selections and improving accuracy. We have described our work to adapt a range of leading 2D pointing assistance techniques into this interesting and increasingly common 3D pointing scenario. Further, we have shown how an adapted Bubble Cursor technique can perform extremely well under a range of target characteristic conditions. We also found that one other technique, Target Gravity, performs particularly well in scenarios with sparse or moving targets, such as in games or UIs (e.g., menus). Our work provides valuable new information that designers of mobile augmented reality can put into practice. Our work illustrates how an area with long-history of HCI research can be leverage to dramatically improve usability in a new context.

Moving forward, we will examine how assistance techniques can be employed in other Augmented Reality scenarios with new devices. In particular, we will explore how to adapt this cursor-based and target assistance approach to head-mounted AR devices, where a fixed-cursor approach many not be the best approach and where distance pointing techniques have not been well researched.

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
