# OpenReview forum: "Assistance for Target Selection in Mobile Augmented Reality"
_graphicsinterface.org/Graphics_Interface/2020/Conference — GI 2020_

### Official Review · AnonReviewer3 · 2020-04-17
**good paper, sound implementation and grounded design choices, a thorough evaluation with outcomes and discussion that may be of interest to the community**

**Rating:** 7
**Confidence:** 3

**Review:**

This paper looks to address the challenge of target selection in mixed and augmented reality applications. Target selection is difficult in MAR because  targets may be moving, occluded, and vary in size based on distance. Providing target assistance to MAR users is proposed as a means to help improve MAR system usability. Five techniques are investigated (baseline, bubble, sticky, gravity, touch). Target assistance is not a novel idea and the authors leverage existing methods that have been applied in cursor-based applications in 2D. These methods are applied with some adaptations.

The paper does a good job at reviewing the related work particularly in cursor-based techniques and 3D. However, I felt that less was discussed about the work in MAR. It also wasn’t too clear on differences with target assistance in 3D games. Making this distinction would be helpful.

A user study is conducted to asses how target assistance techniques may help increase speed and accuracy of selection. For the study task, targets are placed in a diverse type of scenarios e.g. co-planar, varying distances from user, mobile, occluded, etc.

I found the discussion interesting. The authors discuss the strengths and weaknesses of each technique and provide design recommendations for different applications. E.g. Target gravity is recommended in scenarios where selection objects are sparse and bubble cursor may not be as appealing when the visuals may interfere.
How do these findings compare to prior work either in MAR or other contexts? Have these target assistance techniques shown similar strengths weaknesses across comparable scenarios?

Additional Questions:
How was the target size selected?
Did participants walk around during the evaluation?
Beyond applying target assistance techniques that have been shown beneficial in other scenarios, could there be unique ways to leverage the MAR context to reduce the high error rates?

Grammatical errors in:
Typo in last sentence of last paragraph in Related Work > Target Selection in 3D
Typo in first sentence of last paragraph in "Are the different Target Types Scenarios useful? “

Overall, this was a good paper, sound implementation and grounded design choices, a thorough evaluation with outcomes that may be of interest to the community. I recommend for accepting this work.

---

### Official Review · AnonReviewer2 · 2020-04-17
**Nice study, the primary concern was about some strong claims.**

**Rating:** 7
**Confidence:** 3

**Review:**

This submission compared different target selection techniques in mobile augmented reality (AR). It is a timely exploration and evaluation of the choices available on the mobile AR environment.

The paper is overall well-written and easy to read. I especially like the subsection titles in the discussion, which highlight the topic and attract the reader with the most exciting research questions.

However, I do have concerns about some strong claims, as well as some of the study designs.

My primary concern is that, compared to the carefully designed cursor-based techniques, the touch selection in the user study is a complete naive implementation. Some simple "assistance" should be able to improve the performance of the touch condition significantly. For example, instead of only using the single touchpoint for selection, using the contact area (or even a small-sized sphere around the touchpoint) for selection could facilitate the selection in the user study. There are also many advanced techniques for improving touch selection, for example [1,2]. I suggest the authors add a discussion about this limitation, and, more importantly, DO NOT claim touch selection in mobile AR is worse than cursor-based selection. As a matter of face, touch should be faster in "targeting," since the user only needs to put the finger on the screen while moving the cursor requires more body movement in mobile AR.

[1] Wang, F., & Ren, X. (2009, April). Empirical evaluation for finger input properties in multi-touch interaction. In Proceedings of the SIGCHI Conference on Human Factors in Computing Systems (pp. 1063-1072).

[2] Benko, H., Wilson, A. D., & Baudisch, P. (2006, April). Precise selection techniques for multi-touch screens. In Proceedings of the SIGCHI conference on Human Factors in computing systems (pp. 1263-1272).

I am also slightly concerning about the recorded time. During the trails, after a participant answering one trial, the next target will be randomly chosen. If the next target is close to the previous one, then less time is required for moving the cursor.

I am also curious about the physical movement of the participants. As stated by the authors, they allowed participants to move around, but advise them that more time may be required. I consider this as discouraging physical movement, while physical movement is quite essential when using a real-world mobile AR application.

The visual feedback in the bubble cursor condition can be improved. As discussed by the authors, the visual feedback in the bubble cursor is essential for the user to understand the selection area. However, from the provided video, it seems the implementation in this user study did not have the "morphing" feature which is used to contain the target when it is not completely contained by the main bubble.

The authors include a discussion of the pros and cons of each technique in the discussion. However, it is hidden in the text. Since such discussion can be unrelated to the experiment results, the authors can discuss this before introducing the study and give the reader a general sense of the advantages and disadvantages of each technique.

The first three paragraphs in the RELATED WORK section should be moved to the beginning of the discussion section.

I appreciate that the authors tested five scenarios. However, I am also curious if other factors may affect the tested task performance from the literature (e.g., from the VR studies). For example, I can consider the screen size of the mobile devices can be one possible factor. Also, the study only includes only regular layouts of targets, which is impractical. Density may also need to be discussed. I suggest the authors add a discussion of the mentioned limitations in the study design.

Some minor things:
 * The bubble cursor was mentioned many time to be "well-known," I like the technique too, but there is no need to emphasize many times.
 * When removing outliers, were the standard (time above 3 s.d. than the mean) considered within each participant, or across all participants?
 * At the end of the first paragraph in the STUDY section, you mentioned the five scenarios in a different order than anywhere else.

Overall, I think this study provides preliminary insights into object selection techniques in mobile AR. I would argue to accept this submission.

---

### Official Review · AnonReviewer1 · 2020-04-21
**Acceptable, mostly incremental work.**

**Rating:** 7
**Confidence:** 3

**Review:**

Overall, I have very few complaints about this submission: the motivation is convincing, the related work has the proper width and depth, the rationale for choosing selection techniques is reasonable, the experimental setup follows standard procedures, the data analysis seems correct, and the discussion provides some insights.

My first point of criticism is that the work in its current state is mostly incremental. As the authors have pointed out, there has been a plethora of published work in 2D target acquisition (mostly static desktop UIs, highly dynamic video games, etc.), and the results presented in this submission, while confirming previous research, do not go far beyond what we already knew.

My second point of criticism is that the authors are vague about the contribution of their work. I would have generally preferred to see a deeper theory-based discussion on what performance the authors expect using well-established techniques in the different context of MAR. Currently, I feel that the authors approach was “let’s test some selection techniques in MAR and see what happens”; this is not a scientific approach, this is product testing.

I would still argue for accepting this submission for two reasons: first, I believe that this work can be a great first step towards a more detailed analysis of what makes MARs different from desktops and video games (e.g., unlike desktops, they are not static, but unlike video games, only users introduce dynamics). Second, I also believe that the threshold for accepting work at GI is somehow lower than at other conferences (e.g., CHI), and we should accept work, especially if it has the potential to become more impact-full in the future.

---

### Meta-Review · Area_Chair1 · 2020-04-24

**Recommendation:** Accept
**Confidence:** 3

**Metareview:**

Meta by R1:
Assistance for Target Selection in Mobile Augmented Reality - Meta Review

Overall, all three reviewers agree that this submission is at a level that warrants acceptance into GI 2020.

There are three main issues that I believe the authors should improve in future iterations of this draft.
First, they should address the limitations of their work (=> limitation section). The touch-condition was implemented in a way that does not represent the current state-of-the-art (R2). Using such an inferior implementation in a comparative study is bad scientific practise; the rationale for this decision has to be laid out. The study layout actively discourages physical movement of the participants (R2). Again, this is a design decision that has to be better explained, as this implies that the results might change in a different scenario.

Second, the authors should be clearer about their contribution (=> introduction, discussion, and conclusion). While reviewers generally lauded the background research (R1, R3), the contribution was perceived as vague (R1) and some claims as too strong (R2). The authors should be utmost precise about what their work adds to the body of knowledge: "present[ing] the first study" does not warrant publication, and providing "a reference implementation" implies that the authors will provide some source code.

Third, the authors should fix all the minor problems pointed out, particularly by the reviewers (R2, R3).

---

### Decision · Program_Chairs · 2020-04-25

Accept